# Gait Analysis in Children with Cerebral Palsy: Are Plantar Pressure Insoles a Reliable Tool?

**DOI:** 10.3390/s22145234

**Published:** 2022-07-13

**Authors:** Maria Raquel Raposo, Diogo Ricardo, Júlia Teles, António Prieto Veloso, Filipa João

**Affiliations:** 1CIPER, Faculdade de Motricidade Humana, Universidade de Lisboa, Estrada da Costa, Cruz-Quebrada-Dafundo, 1499-002 Lisbon, Portugal; raquel.braposo@scml.pt (M.R.R.); diogo.ricardo@estesl.ipl.pt (D.R.); jteles@fmh.ulisboa.pt (J.T.); apveloso@fmh.ulisboa.pt (A.P.V.); 2Escola Superior de Tecnologia da Saúde de Lisboa (ESTeSL), Instituto Politécnico de Lisboa, Av. D. João II, 1990-096 Lisbon, Portugal

**Keywords:** plantar pressure, cerebral palsy, gait analysis, reliability, insoles

## Abstract

Cerebral palsy (CP) is a common cause of motor disability, and pedobarography is a useful, non-invasive, portable, and accessible tool; is easy to use in a clinical setting; and can provide plenty of information about foot–soil interaction and gait deviations. The reliability of this method in children with CP is lacking. The aim of this study is to investigate test–retest reliability and minimal detectable change (MDC) of plantar pressure insole variables in children with CP. Eight children performed two trials 8 ± 2.5 days apart, using foot insoles to collect plantar pressure data. Whole and segmented foot measurements were analyzed using intraclass correlation coefficients (ICC). The variability of the data was measured by calculating the standard error of measurement (SEM) and the MDC/ICC values demonstrated high test–retest reliability for most variables, ranging from good to excellent (ICC ≥ 0.60). The SEM and the MDC values were considered low for the different variables. The variability observed between sessions may be attributed to the heterogeneous sub-diagnosis of CP.

## 1. Introduction

Cerebral palsy (CP) is the most common cause of motor disability in children [1,2,3]. CP is a complex pathology that describes a group of impairments and motor disorders, which are permanent but not immutable, resulting from a nonprogressive cerebral disorder [4] with different presentations and functional levels [5].

CP presents both positive features such as spasticity, hyper-reflexia, and co-contraction, and negative features including weakness, difficulties in motor control, and sensory and balance impairments [6]. The lack of control is obvious at the lower limb joints, especially the ankle joint. These alterations are the main cause of limb contractures, musculoskeletal deformity, and gait deviations [7].

Foot deformities, along with hip displacement, are the most common musculoskeletal occurrences in CP. Among the most common foot deformities in this population are equinus, planovalgus, and equinovarus, which can vary from very mild and flexible to severe and rigid [8]. These deformities, which cause the foot to abnormally lay on the ground, can significantly impair function and quality of life; however, very few studies have systematically investigated the foot morphology and the ground–foot interaction during the stance phase in this population [7].

Instrumented clinical gait analysis has been an excellent tool for planning intervention and assessing outcomes in the rehabilitation process of children with CP [1,2]. Though the gold standard for gait analysis in children with CP would be a quantitative three-dimensional analysis of movement and respective articular moments and power (kinematics and kinetics), possibly alongside muscle activation (electromyography) and oxygen consumption [9], it is not always possible to conduct such an assessment in a clinical setting. More accessible and portable methods have been recently used such as inertial sensors [10,11] and plantar pressure recording devices [7,12,13,14,15].

Under this aspect, dynamic pedobarography is a relatively simple, portable, and non-invasive technology that measures the change in plantar pressure distribution throughout the stance phase of gait [16]. It is an easy method to use in a clinical setting; can provide plenty of information about foot–soil interaction; and, alongside other gait analysis methods, can help assess the impact of a medical intervention, a rehabilitation program, or the effects of an orthotic device. Several studies tested its reliability [16,17] for both healthy adult and children, but none have assessed subjects with CP. The few existing clinical studies in participants with CP use mainly plantar pressure mats/platforms instead of insoles [7,12,13,14,15].

In the past years, several studies have tried to produce normative age-dependent gait databases [18,19,20], which are fundamental to assess and compare with pathologic situations. In fact, more evidence is now surfacing about the foot characteristics of typically developed children. Foot pressure changes dramatically throughout the life cycle, especially in the early years (up to 6 years old). The evidence shows that, while younger typically developing children present with a flatfoot pattern, older children tend to develop a more curvilinear pattern [18]. Moreover, older children show greater values in the main plantar pressure variables when compared with younger children [20].

Even fewer studies have included plantar pressure measurements in children with CP. There has been no attempt to create any kind of database, which is fundamental to assess and compare the natural progression of the condition and the results of medical and therapeutic interventions. Nevertheless, data collected across the existing studies show that there is a variability in foot pressure distribution depending on spasticity overall, there is an increase in pressure towards the toes and forefoot as well as a significant reduction towards the heel [7,14,21].

Reports of plantar pressure data in the literature are highly heterogeneous. One of the challenges of standardizing this tool is that there are multiple footprint segmentation models [19]. There is still no consensus about which foot model may provide the most detailed information, without losing the functional aspects of the foot [15]. Most authors propose an anatomical/functional segmentation, corresponding to the foot joint positions, which ranges from as few as 3 to as many as 12 subdivisions of the footprint (the most often used are the hind-foot, mid-foot (medial and lateral), forefoot (medial and lateral), and toes (toes 2–5 and the first toe) [7,13,14,15,17,18,19,21,22,23,24].

The absence of systematized evidence regarding the reliability of foot pressure insoles on this specific population and the need to assess the dimension of error measurement with this tool calls for further investigation. In so, the aim of this study is to investigate test–retest reliability and minimal detectable change of plantar pressure insoles in a sample of children with CP when walking in regular footwear.

## 2. Materials and Methods

### 2.1. Design

Prospective intra-rater test–retest reliability and minimal detectable change study.

### 2.2. Participants Selection

A convenience sample of 10 children with cerebral palsy was selected from a Portuguese rehabilitation center to participate in this study. The selected participants followed the eligibility criteria: male or female children between 4 and 12 years of age, foot length ranging from 15 to 20 cm (because of equipment constraints), with a clinical diagnosis of bilateral (lower limb predominance) or unilateral cerebral palsy, grades I and II on the Gross Motor Function Classification System (GMFCS) [25], able to walk independently for 5 m without walking aids, and able to comprehend and comply with simple instructions. Children should also have not been subjected to orthopedic surgery or botulinum toxin treatment in the previous 6 months. The protocol was approved and executed in accordance with the Faculty of Human Kinetics Ethics Committee (CEFMH-2/2019). All procedures were previously explained to both the child and the legal guardian, an informed consent form was filled and signed by the legal guardian, and verbal consent was given by the child.

### 2.3. Data Collection Protocol

Data collection was performed on two different days within a period of 7 to 14 days (8 ± 2.5 days) to minimize the assessor memory bias and to prevent a change in the children’s gait pattern or clinical condition. Clinical history and a brief physical exam (mass, height, lower limb posture, selective motor control tests, gastrocnemius length, and spasticity) [9] were conducted in the first session.

Children wore the foot insoles Pedar-X system^®^ (Novel, Munich, Germany), inside their usual footwear (adequate to their feet size) and no socks. The children wore the same pair of shoes for both trials. The batteries and the wireless transmitter were strapped or placed inside a backpack on the child’s back. A schematic picture and a photograph illustrate the experimental setup used (Figure 1). The insoles were calibrated using the Pedar X Standard (v 25.3.6, Novel, Munich, Germany) protocol (before the beginning of each trial, the participant was asked to lift one foot at a time off the ground for approximately 15 s). Data were sampled at 100 Hz. Children were instructed to walk back and forth, along a 5 m line drawn on a smooth and regular floor, unassisted and at a self-selected speed, without running. A chair was placed at either end of the walkway, in case the participants needed to stop. Data collection stopped after 2 min if the children achieved a minimum of 15 steps with each lower limb.

### 2.4. Data Processing

Data were extracted and processed using the Novel Multiprojects-e (v 24.3.34, Novel, Munich, Germany), which enabled the creation of a database and processing of each participant’s individual footprint. Each data set was reviewed and amiss footprints and directional changes were wiped out of the original records. The average of the selected variables (force–time integral, pressure–time integral, maximum force, peak pressure, contact area, and contact time) was automatically calculated by the software for the whole foot. A mask then divided the foot into three regions (hindfoot, middle foot, and forefoot), according to the length of the foot (0 to 30%, 30 to 60%, and 60 to 100% of total length, respectively), as shown in Figure 2. These masks were applied automatically by the software, and average scores were calculated for each variable and zone. The software also produced 3D plantar pressure maps for each participant, allowing a visual comparison of the first and second trial (Figure 3).

### 2.5. Statistical Analysis

Statistical analysis to assess the test–retest reliability of plantar pressure data was carried out using the methodology described by Koo and Li (2015) [26], similar to the methods used by Fernandes et al. (2015) [27] and Ricardo et al. (2021) [28] in their works.

Intraclass correlation coefficients (ICCs) considering the two-way mixed model with absolute agreement and accounting for the mean of multiple measurements [26] were calculated for all variables and masks, and a critical level of *p* < 0.05 was considered significant. The *ICC* statistical analysis was performed using SPSS (version 28.0.0; IBM, Chicago, IL, USA), using the following formula:$$ICC=\frac{MS_{R}-MS_{E}}{MS_{R}+\frac{MS_{C}-MS_{E}}{n}}$$
where *MS_R_* represents the mean square between lower limbs; *MS_E_* represents the mean square for error; *MS_C_* represents the mean square within lower limbs, concerning the selected pedobariografic variables; and *n* is the total number of lower limbs assessed (two lower limbs for each of the eight participants).The level of agreement was considered poor, fair, good, and excellent when *ICC* < 0.40, 0.40 ≤ *ICC* < 0.60, 0.60 ≤ *ICC* < 0.75, and 0.75 ≤ *ICC* ≤ 1.00, respectively [29]. Calculations also included the mean difference between measurements (Mean_diff_), the 95% CI for the Mean_diff_, the standard deviation of the differences (*SD_diff_*), and the 95% Bland and Altman limits of agreement (95% LOA).

The absolute measure of reliability standard error of measurement (*SEM*) was calculated using the following equation:$$SEM=\frac{SD_{diff}}{\sqrt{2}}$$
where *SD_diff_* represents the standard deviation of the difference.

To determine the smallest amount of change that must be achieved to reflect a true change, outside the error of the tests, the minimal detectable change (*MDC*) was calculated using the following equation:$$MDC=1.96\cdot \sqrt{2}\cdot SEM$$

The *SEM* and *MDC* were calculated using Microsoft Excel 2013 (Microsoft Corporation, Redmond, WA, USA).

## 3. Results

The participants of the study were a convenience sampling of ten children with CP (nine spastic unilateral, one spastic bilateral; four females, six males; age 57.9 ± 13.4 months; height 110.4 ± 7.6 cm; mass 18.1 ± 2.4 kg) (Table 1), two of which dropped out of the study as they could not complete the trials in the same time frame as the other participants (one because of COVID-19 prophylactic quarantine and the other because of loss of contact). Data from each limb were processed separately (N = 16), because of the heterogeneous physical presentation of unilateral CP that composed most of the selected sample. On average, we assessed 75.8 ± 27.9 steps on each trial.

### 3.1. Reliability of Whole Foot Measurements

As shown in Table 2, all selected variables calculated for the whole footprint showed an excellent ICC (ICC ≥ 0.75), except for the contact time variable (ICC = 0.36, 95% CI 0 to 0.784). The SEM and MDC values were within an acceptable range for each of the variables.

### 3.2. Reliability of Segmented Foot Measurements

Overall ICC values for the segmented foot measurements fit in the good to excellent range (ICC values ≥ 0.60), except for peak pressure (ICC = 0.439, 95% CI 0 to 0.807) and maximum force (ICC = 0.552, 95% CI 0 to 0.845) at the forefoot (Table 3). The SEM and MDC values were within an acceptable range for each of the variables.

## 4. Discussion

The main objective of the current study was to assess the intersession and intra-rater reliability of plantar pressure variables when using pressure foot insoles and, to the best of the authors’ knowledge, it is the first study to do so. Plantar-pressure-related data for children with CP are still scarce in published evidence. Alongside other gait analysis tools, pedobarographic measurements are useful in assessing pre- and post-surgical outcomes, treatment with botulinum toxin, and orthotic management, as they provide important information about foot pressure distribution, postural control, center of pressure (COP) displacement, and the foot–soil interaction. Nonetheless, if this type of data is to be used for assessing clinical or therapeutic interventions, it is of high importance to establish reliability levels for this specific method and population [24].

The reliability of foot pressure platforms or mats for typically developing children and healthy adults has been previously established by Cousins et al. (2012) [33], Hafer et al. (2013) [16], and Niller et al. (2016) [17]. Other similar studies assessed likewise reliability for both typically developing children and children with CP, also using a plantar pressure mat [14,34]. However, the use of plantar pressure foot insoles presents with different benefits, such as the possibility of their use inside shoes or orthotic devices recording a higher number of gait cycles, as well as overall being easier to use with smaller children.

Our results show high reliability (ICC ≥ 0.60) for 21 of the 24 parameters that were tested. Still, three of the outcome measures for whole foot and forefoot showed lower values (whole foot contact time variable and peak pressure and contact time variables at the forefoot).

The number of participants included in this study was small, but similar to other researches [14,22]. However, because of the heterogeneity of children with CP, we opted to conduct a separate analysis of right and left feet. This increases the total sample to sixteen (feet). Post-hoc power analysis with α = 0.05 revealed good power (≥0.90) for most variables, except for the three variables mentioned above. Post-hoc statistical analysis was carried out using R software (version 4.1.3., R Core Team 2022) [35] and the “ICC.Sample.Size” package (version 1.0.) [36].

The poor reliability results for the contact time variable (whole foot and forefoot region) may be explained by the heterogeneous gait pattern with which the participants presented. Most of our sample were children with unilateral CP, who present with a slower pace and abnormal weight shift between the affect side and less affected size. As a separate limb analysis was conducted, the diminished weight shift to the more affected side may have led to an increased contact time on the opposite side, and thus the contact time variable registered a wider range of values. Moreover, although we asked the children to walk at a self-selected comfortable pace, their pace varied.

The lower ICC values obtained from the forefoot peak pressure can be attributed to the slight discrepancy between the total foot length and the length of the available insole. Foot length across our sample ranges from 15 cm to 20 cm, but the same pair of 20 cm insoles was used throughout the investigation. This means that the fit was not always perfect, leaving vacant pressure cells at the top of the insoles, which can reflect in the forefoot values. Moreover, the total weight of the equipment was 0.5 kg, which may impact the trials of some of the smaller children and those with greater locomotion difficulties and gait deviations.

The *SEM* and *MDC* values were determined to quantify the amount of error associated with each variable in this population. Even though the *SEM* and *MDC* values for each variable showed a clinically acceptable level of error [20], they were transformed into a percentage for comparison purposes:$$SEM\%=\frac{SEM}{Mean}\cdot 100$$

And
$$MDC\%=\frac{MDC}{Mean}\cdot 100$$

Please refer to Ayán-pérez, C. and Bouzas-rico, S. (2019) [37] for more information. For reference purposes, *MDC*% scores >30% were considered poor, from 10 to 30% were considered acceptable, and <10% were considered excellent [38]. The obtained values for *MDC*% were all considered to be poor, except for the contact area variable for the whole foot and peak pressure and contact area for the midfoot, which were within the acceptable range. These results are equivalent to other similar studies [37,39].

Various foot segmentation models have been reported in recent literature [7,13,14,15,17,18,19,21,22,23,24]. Complex masking usually involves anatomical and functional segmentation, including external references (for example, retroreflective markers and an optoelectronic system) that were not available for this specific study. Smaller areas of division may provide with less detailed information, and they are also more error-prone [17]. A three identical part division masking was selected for this study, similar to that of Galli et al. [7], allowing to differentiate force, pressure, and spatio-temporal values between the hind-foot, midfoot, and forefoot. Knowing that most participants presented an equinus gait pattern, we expected altered values in these three areas, and that division allowed the retrieval of more specific data.

The absence of previous reliability studies with this population and method precludes comparisons with similar *SEM* and *MDC* data. These preliminary results could prove useful to determine clinical changes in foot pressure and understand how those changes differentiate from the error of measurement. This is particularly important in studies where we have a pre- and post-assessment of the participant to see the effect of an intervention process. If the post results are superior to the reported error of the measurement, we can be confident in stating that there was a significant effect caused by the intervention.

## 5. Conclusions

This study is the first that establishes plantar pressure insoles as a reliable tool for measuring different gait-related variables in children with CP. The results indicate a good reliability for most variables, except for whole foot contact time and peak pressure and contact time at the forefoot. These lower values observed may be attributed to the heterogeneous gait pattern of children with CP and the above-mentioned equipment limitations of the study.

## Figures and Tables

**Figure 1 sensors-22-05234-f001:**
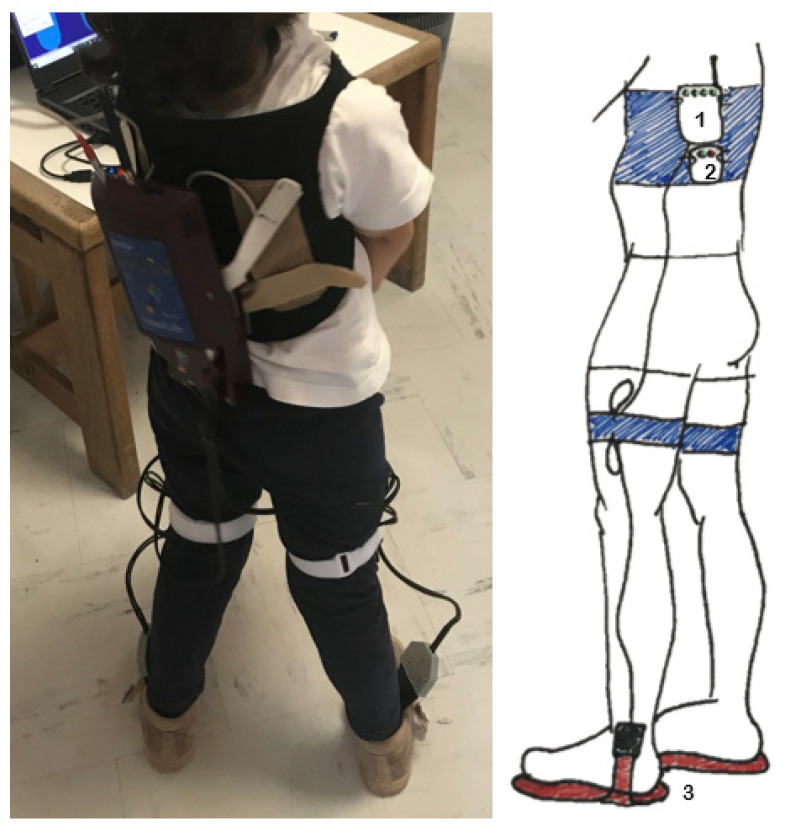
Experimental set up (1—wireless transmitter; 2—batteries; 3—plantar pressure insoles).

**Figure 2 sensors-22-05234-f002:**
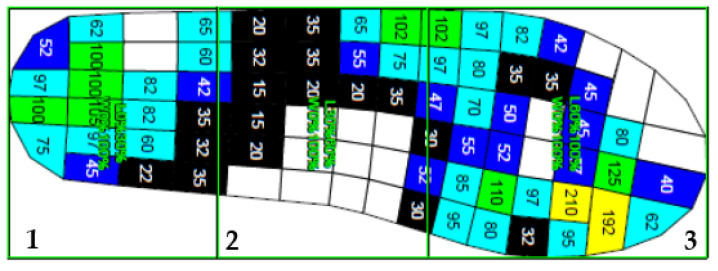
Three zones of segmentation of the foot (1—0% to 30% of total length; 2—30% to 60% of total length; 3—60% to 100% of total length). Obtained from Novel Multiprojects-e (v 24.3.34, Novel, Munich, Germany).

**Figure 3 sensors-22-05234-f003:**
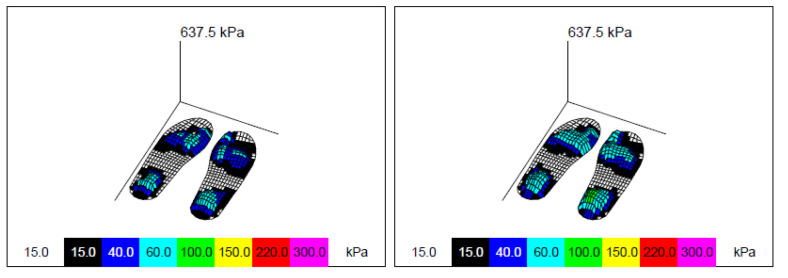
Three-dimensional plantar pressure mapping for test–retest results of participant 008. Obtained from Novel Multiprojects-e (v 24.3.34, Novel, Munich, Germany).

**Table 1 sensors-22-05234-t001:** Participants’ characteristics.

Participant	Gender	Age (Months)	Diagnosis	Affected Side	GMFCS Level [25]	Interval between Trials (Days)	Mass (kg)	Height (cm)	Sagittal Gait Pattern [30,31]		Gastrocnemius Spasticity (Modified Ashworth Scale) [32]		Foot Length (cm)		Number of Steps (Average from Both Trials)		Status
									Right	Left	Right	Left	Right	Left	Right	Left	
001	Male	54	Unilateral CP	Right	I	14	16.5	105	Drop Foot	-	1	0	15	16	70	70	Completed trials
002	Male	65	Unilateral CP	Left	II	9	20	118	-	True Equinus	0	4	19	17	52	59	Completed trials
003	Female	41	Unilateral CP	Right	II	7	19	105	True Equinus	-	1+	0	16	17	55	52	Completed trials
004	Female	56	Bilateral CP	Both	II	7	18	110	Apparent Equinus	Apparent Equinus	1	1	17	17	55	55	Completed trials
005	Female	65	Unilateral CP	Right	I	7	20.4	120	True Equinus	-	1	0	20	19	64	65	Completed trials
006	Male	45	Unilateral CP	Left	I	-	13	97	-	True Equinus	0	1	15	15	-		Dropped out
007	Male	41	Unilateral CP	Right	I	-	16	103	True Equinus	-	1	0	15	16	-		Dropped out
008	Male	74	Unilateral CP	Right	I	7	20.5	115	True Equinus	-	1	0	20	20	75	74	Completed trials
009	Male	80	Unilateral CP	Right	I	6	20.1	115	True Equinus	-	1+	0	19	19	122	120	Completed trials
010	Female	58	Unilateral CP	Right	I	7	17	116	Equinus/Jump Knee	-	2	0	16	18	112	119	Completed trials

**Table 2 sensors-22-05234-t002:** Reliability values for pedobarography measurements (whole foot).

Pedobarograpy Measurements	ICC	ICC 95% CI	Mean	Mean Diff	SD Diff	95% LOA	SEM	MDC
Force–time integral (N·s)	0.76	(0.30; 0.92)	73.72	−2.08	18.57	(−38.47; 34.31)	13.13	36.39
Pressure–time integral (kPa·s)	0.89	(0.70; 0.96)	55.40	0.63	10.04	(−19.05; 20.31)	7.10	19.68
Maximum force (N)	0.79	(0.42; 0.93)	161.30	−7.61	25.00	(−56.61; 41.40)	17.68	49.00
Peak pressure (kPa)	0.81	(0.47; 0.93)	136.45	6.84	27.48	(−47.01; 60.70)	19.43	53.85
Contact area (cm^2^)	0.83	(0.53; 0.94)	56.80	−3.69	8.15	(−19.66; 12.27)	5.76	15.97
Contact time (ms)	0.37	(0; 0.78)	669.93	4.29	137.30	(−264.81; 273.40)	97.08	269.11

**Table 3 sensors-22-05234-t003:** Reliability values for pedobarography measurements (three zones of the segmented foot).

	Pedobarograpy Measurements	ICC	ICC 95% CI	Mean	Mean Diff	SD Diff	95% LOA	SEM	MDC
Hindfoot	Force–time integral (N·s)	0.83	(0.51; 0.94)	17.44	−1.43	11.35	(−23.67; 20.82)	8.02	22.24
	Pressure–time integral (kPa·s)	0.97	(0.92; 0.99)	21.41	0.62	12.01	(−22.93; 24.16)	8.49	23.54
	Maximum force (N)	0.92	(0.77; 0.97)	70.50	−6.38	28.65	(−62.53; 49.77)	20.26	56.15
	Peak pressure (kPa)	0.88	(0.65; 0.96)	78.56	−3.84	18.48	(−40.06; 32.37)	13.07	36.22
	Contact area (cm^2^)	0.91	(0.75; 0.97)	13.68	−1.76	6.19	(−13.89; 10.36)	4.38	12.13
	Contact time (ms)	0.86	(0.62; 0.95)	365.79	38.16	272.29	(−495.53; 571.85)	192.54	533.69
Middle Foot	Force–time integral (N·s)	0.91	(0.75; 0.97)	15.32	0.52	3.14	(−5.63; 6.67)	2.22	6.15
	Pressure–time integral (kPa·s)	0.97	(0.92; 0.99)	30.19	0.91	5.95	(−10.75; 12.57)	4.21	11.66
	Maximum force (N)	0.91	(0.74; 0.97)	47.92	−2.32	7.84	(−17.69; 13.05)	5.54	15.37
	Peak pressure (kPa)	0.97	(0.92; 0.99)	74.89	1.19	8.31	(−15.09; 17.47)	5.87	16.28
	Contact area (cm^2^)	0.98	(0.94; 0.99)	16.54	−0.34	2.07	(−4.39; 3.72)	1.46	4.06
	Contact time (ms)	0.73	(0.25; 0.90)	621.32	9.79	118.82	(−223.09; 242.67)	84.02	232.88
Forefoot	Force–time integral (N·s)	0.73	(0.25; 0.90)	40.95	−1.18	11.14	(−23.02; 20.66)	7.88	21.84
	Pressure–time integral (kPa·s)	0.97	(0.92; 0.99)	42.35	1.53	7.30	(−12.77; 15.83)	5.16	14.30
	Maximum force (N)	0.73	(0.26; 0.90)	123.44	−5.93	23.40	(−51.80; 39.95)	16.55	45.87
	Peak pressure (kPa)	0.44	(0; 0.81)	124.59	8.68	28.00	(−46.19; 63.55)	19.80	54.87
	Contact area (cm^2^)	0.68	(0.07; 0.89)	25.59	−3.57	7.21	(−17.70; 10.55)	5.10	14.12
	Contact time (ms)	0.55	(0; 0.85)	578.39	22.66	194.57	(−358.70; 404.02)	137.58	381.36

## Data Availability

The data presented in this study are available on request from the corresponding author. The data are not publicly available due to privacy issues.

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
