# Peer review of "Gait Analysis in Children with Cerebral Palsy: Are Plantar Pressure Insoles a Reliable Tool?"

_sensors, 2022, doi:10.3390/s22145234_

Round 1
Reviewer 1 Report
Well written article. The gold standard method for testing CP patients would be motion capture systems but pressure measuring could be a useful tool when custom footwear is made.
I would only argue that there were may still not enough participants recruited for this research.
From the one hand this research does not give anything new or anything useful from another hand if the question is asked how reliable plantar pressure to be used in footwear research for specifically CP patients and this would be the article.
Could be one main issue not enough people but author argued that there were number of similar research with the same number of subjects. But I still doubt the reliability question can be answered with 7 subjects.
The topic is relevant to the field, however plantar pressure is not widely used for CP patients testing. Motion capture labs are the main tool for that. However, there were studies like custom shoes on insoles testing with plantar pressure systems. https://jfootankleres.biomedcentral.com/articles/10.1186/s13047-016-0154-5
Authers said that there was not any research done if plantar pressure were reliable for CP patients walking in shoes. There is one similar article plantar pressure used for postural control and 112 subjects were recruited. http://old.scielo.br/scielo.php?script=sci_arttext&pid=S1980-65742020000300304&lng=en&nrm=iso&tlng=en
There is another similar research that tested barefoot reliability where 45 CP kids were recruited. https://jfootankleres.biomedcentral.com/articles/10.1186/1757-1146-5-8
I would say that they answered that question, but this is just extra with limited of subjects.
Methodology is good however I would suggest recruiting more subjects as gait deviation in CP patients can be very vary. To answer the article main question how reliable plantar pressure for CP patients with 7 subjects is questionable. I test around 10 CP patients with motion capture system every month and results for kinematic data would be different for some joints and planes. I would also suggest adding that the same shoe was used for each subject, and they did not change them during that research, or did they?
Evidence suggest that overall plantar pressure is reliable tool for CP patients and the main questions was answered. Would be good if some new research more and more used this method by comparing custom made footwear profiles and insoles for CP patients and see the plantar pressure effect. But gait laboratory with force platforms, motion capture system and EMG is a gold standard. Plantar pressure analysis is not that developed or widely used to provide valuable information about walking gait alterations. However, may be good for comparing custom orthopedic devices.
Not much work done in that area and there are some old references and few new ones. Authors could add more articles like that few articles are published which used plantar pressure for postural and stability assessment. But not necessary https://www.mdpi.com/2075-4426/12/3/394/htm
Could be added that barefoot plantar pressure was tested
The reliability of plantar pressure assessment during barefoot level walking in children aged 7-11 years
https://www.researchgate.net/publication/221876839_The_reliability_of_plantar_pressure_assessment_during_barefoot_level_walking_in_children_aged_7-11_years
Tables and figure are clear for me. But would be good to add at least one comparison image of 3D plantar pressure readings before and after 7-14 days.
Author Response
Well written article. The gold standard method for testing CP patients would be motion capture systems but pressure measuring could be a useful tool when custom footwear is made.
I would only argue that there were may still not enough participants recruited for this research.From the one hand this research does not give anything new or anything useful from another hand if the question is asked how reliable plantar pressure to be used in footwear research for specifically CP patients and this would be the article. Could be one main issue not enough people but author argued that there were number of similar research with the same number of subjects. But I still doubt the reliability question can be answered with 7 subjects.
The topic is relevant to the field, however plantar pressure is not widely used for CP patients testing. Motion capture labs are the main tool for that. However, there were studies like custom shoes on insoles testing with plantar pressure systems. https://jfootankleres.biomedcentral.com/articles/10.1186/s13047-016-0154-5
Authers said that there was not any research done if plantar pressure were reliable for CP patients walking in shoes. There is one similar article plantar pressure used for postural control and 112 subjects were recruited. http://old.scielo.br/scielo.php?script=sci_arttext&pid=S1980-65742020000300304&lng=en&nrm=iso&tlng=en
There is another similar research that tested barefoot reliability where 45 CP kids were recruited. https://jfootankleres.biomedcentral.com/articles/10.1186/1757-1146-5-8
I would say that they answered that question, but this is just extra with limited of subjects.
Methodology is good however I would suggest recruiting more subjects as gait deviation in CP patients can be very vary. To answer the article main question how reliable plantar pressure for CP patients with 7 subjects is questionable. I test around 10 CP patients with motion capture system every month and results for kinematic data would be different for some joints and planes. I would also suggest adding that the same shoe was used for each subject, and they did not change them during that research, or did they?
Evidence suggest that overall plantar pressure is reliable tool for CP patients and the main questions was answered. Would be good if some new research more and more used this method by comparing custom made footwear profiles and insoles for CP patients and see the plantar pressure effect. But gait laboratory with force platforms, motion capture system and EMG is a gold standard. Plantar pressure analysis is not that developed or widely used to provide valuable information about walking gait alterations. However, may be good for comparing custom orthopedic devices.
Not much work done in that area and there are some old references and few new ones. Authors could add more articles like that few articles are published which used plantar pressure for postural and stability assessment. But not necessary https://www.mdpi.com/2075-4426/12/3/394/htm Could be added that barefoot plantar pressure was tested
The reliability of plantar pressure assessment during barefoot level walking in children aged 7-11 years
https://www.researchgate.net/publication/221876839_The_reliability_of_plantar_pressure_assessment_during_barefoot_level_walking_in_children_aged_7-11_years
Tables and figure are clear for me. But would be good to add at least one comparison image of 3D plantar pressure readings before and after 7-14 days.
Response: Thank you for your appraisal. We do agree that the gold standard for gait assessment of CP children, is motion capture through the analysis of kinematic and kinetic parameters. As you also mentioned, pedobarography insoles are a useful, non-invasive, portable and accessible tool, easy to use in a clinical setting, which can provide plenty of information about foot-soil interaction and help assess the impact of a medical intervention, a rehabilitation program or the effects of an orthotic device. In this particular study, we intended to highlight the importance of collecting reliable plantar pressure variables, as one type of variables that are relevant to collect in clinical gait analysis.
We also acknowledge that the number of participants is low, but we opted to process the individuals’ lower limbs separately, raising the total sample to sixteen. Also post-hoc power analysis revealed power above 0.90 on most variables. Still our total number of participants does not significantly differ from that of other similar published studies ( Chang et al, 2015; Riad et al. 2007; Ricardo et al., 2021; Stebbins et al, 2005 , all cited in the main article).
We also would like to thank you for the references you sent. We carefully reviewed the studies you suggested. Although the studies refer to the reliability of plantar pressure insoles in healthy subjects, we have included some of its considerations in the Introduction and Discussion sections. Please refer to page 2 lines 47 and 55 and page 11 lines 216, 228 and 239 of the revised manuscript.
We would also like to clarify that the participants’ shoes wore in both trials were the same and also were their usual footwear. Please refer to page 3 line 115 of the revised manuscript.
Lastly, we thank you for the suggestion, and we have added new figures and schematic drawings of the experimental set up and figures to illustrate the 3D plantar pressure results and measurement process. Please refer to page 4 line 126 and page 5 line 146 of the revised manuscript.
We thank you for all your comments. We sincerely hope that the modifications made to our original manuscript meet the concerns you have raised.

Reviewer 2 Report
1. Improve and enhance the presentation of the main idea of the article (chapter Introduction).
The authors should present methods of gait analysis more extensively in the Introduction. It is not only possible to assess the condition of a patient with CP and the improvement of health after rehabilitation by analysing the distribution of foot pressure on the ground (special orthotic insole). In the introduction it is suggested to include a brief presentation of other, alternative studies such as in the paper:
"Inertial Sensors and Wavelets Analysis as a Tool for Pathological Gait Identification".
and a justification as to why pressure distribution testing, this particular method, is chosen as better?
2. Improve the methodology section (Materials and Methods chapter, especially Statistical Analysis).
It is true that the authors cite the sources from which the methodology was adopted, but it is presented in a limited scope and raises many questions. First of all, there is no justification for the choice of Two-Way Mixed-Effects Model and next definition selection Absolute Agreement and the type i.e. single rater or multiple raters. In this case the work:
,,A Guideline of Selecting and Reporting Intraclass Correlation Coefficients for Reliability Research" could be useful and should be cited.
Hence, the formula for calculating Intraclass Correlation Coefficient (ICC) values should appear in the work.
At this point, I would like to draw your attention to correct the editing of the other formulas appearing in the paper in the Statistical Analysis subsection.
3. Correct the Discussion and Conclusion.
a. Detail the very general statement:
"Alongside with other gait analysis tools, pedo-barographic measurements are useful in assessing pre and post-surgical outcomes, treatment with botulinum toxin and orthotic management, as they provide important information about postural alignment, spasticity and muscle weakness and its impact in foot function."
Hence the suggestion to cite other gait testing methods at the beginning of this review.
b. To rethink the discussion of the results and the final conclusion.
The authors write very generally which is questionable due to the small study sample that:
"Our results show high reliability for most parameters.
and immediately afterwards they reflect by writing:
"Still, some outcome measures and foot regions showed lower values (whole foot contact time variable and of peak pressure and contact time variables at the forefoot)."
So it is not: high reliability for most parameters (how many parameters?).
They then very ambiguously try to justify the lower ICC values, as follows:
"The poor reliability results for the whole foot contact time variable may be explained due to the high variability of the participants, whom have a slower pace and an abnormal weight shift than their typical developed peers."
Whereas in Conclusion they write:
"These lower values observed may be attributed to the variability between sessions, as children with CP often show a highly heterogeneous gait pattern."
So what is the final conclusion?
Hence the suggestion to rethink the content of these two chapters, indicating the final conclusions and dividing what is a discussion and what is a conclusion.
Author Response
Point 1: Improve and enhance the presentation of the main idea of the article (chapter Introduction).
The authors should present methods of gait analysis more extensively in the Introduction. It is not only possible to assess the condition of a patient with CP and the improvement of health after rehabilitation by analysing the distribution of foot pressure on the ground (special orthotic insole). In the introduction it is suggested to include a brief presentation of other, alternative studies such as in the paper:
"Inertial Sensors and Wavelets Analysis as a Tool for Pathological Gait Identification".
and a justification as to why pressure distribution testing, this particular method, is chosen as better?
Response 1: Thank you for your suggestions. We do agree with you, pedobarography is not the only nor the better method to assess the gait of CP children. In fact, the gold standard for gait assessment of CP children, is motion capture through the analysis of kinematic and kinetic parameters. However, pedobarography insoles are still a useful, non-invasive, portable and accessible tool, easy to use in a clinical setting, which can provide plenty of information about foot-soil interaction and help assess the impact of a medical intervention, a rehabilitation program or the effects of an orthotic device. We have reviewed the Introduction section and clarified our aim. Please refer to page 2 lines 47 and 55 of the revised manuscript.
Point 2: Improve the methodology section (Materials and Methods chapter, especially Statistical Analysis).
It is true that the authors cite the sources from which the methodology was adopted, but it is presented in a limited scope and raises many questions. First of all, there is no justification for the choice of Two-Way Mixed-Effects Model and next definition selection Absolute Agreement and the type i.e. single rater or multiple raters. In this case the work:
,,A Guideline of Selecting and Reporting Intraclass Correlation Coefficients for Reliability Research" could be useful and should be cited.
Hence, the formula for calculating Intraclass Correlation Coefficient (ICC) values should appear in the work.
At this point, I would like to draw your attention to correct the editing of the other formulas appearing in the paper in the Statistical Analysis subsection.
Response 2: Thank for your comments. We have read the suggested paper and incorporated that reference into our work. Please refer to page 5 lines 152, 155 and 159 of the revised manuscript. We also edited our formulas as you can see in page 5 lines 169 and 175 of the revised manuscript.
Point 3a: Correct the Discussion and Conclusion.
Detail the very general statement:
"Alongside with other gait analysis tools, pedo-barographic measurements are useful in assessing pre and post-surgical outcomes, treatment with botulinum toxin and orthotic management, as they provide important information about postural alignment, spasticity and muscle weakness and its impact in foot function."
Hence the suggestion to cite other gait testing methods at the beginning of this review.
Response 3a: Thank you for your suggestion. As referred in response 1, we have reviewed the gait analysis methods in the Introduction section (page 2 lines 47 and 55 of the revised manuscript). We also clarified the cited sentence. Please refer to page 11 line 216 of the revised manuscript.
Point 3b: To rethink the discussion of the results and the final conclusion.
The authors write very generally which is questionable due to the small study sample that:
"Our results show high reliability for most parameters.
and immediately afterwards they reflect by writing:
"Still, some outcome measures and foot regions showed lower values (whole foot contact time variable and of peak pressure and contact time variables at the forefoot)."
So it is not: high reliability for most parameters (how many parameters?).
They then very ambiguously try to justify the lower ICC values, as follows:
"The poor reliability results for the whole foot contact time variable may be explained due to the high variability of the participants, whom have a slower pace and an abnormal weight shift than their typical developed peers."
Whereas in Conclusion they write:
"These lower values observed may be attributed to the variability between sessions, as children with CP often show a highly heterogeneous gait pattern."
So what is the final conclusion?
Hence the suggestion to rethink the content of these two chapters, indicating the final conclusions and dividing what is a discussion and what is a conclusion.
Response 3b: Thank you for your comments. 21 of the 24 parameters tested showed good reliability results (ICC ≥ 0.60), with the exception of whole foot contact time variable, peak pressure and contact time variables at the forefoot. We have rewritten both the Discussion and Conclusion sections to increase transparency and objectivity of our findings. Please refer to page 11 lines 228 and 239 and page 12 line 287 of the revised manuscript.
We thank you for all your comments. We sincerely hope that the modifications made to our original manuscript meet the concerns you raised.

Reviewer 3 Report
The study presented in the submitted manuscript is interesting, the plantar pressure insole measurement in children with CP. Even though the N is relatively small.
To fit the scope of the Journal, the reviewer suggested the manuscript be revised because the manuscript style is more like a pre-clinical study (with a small population of the sample).
1. Please add a photo or schematic diagram of the measurement process, even though a commercial device may be involved in the measurement. This context will be an exciting part for most readers of Sensors. Please note that the presented photo should not disclose the patient's identity.
2. For the comparative study mentioned in plantar measurements literature in the discussion, please summarize the results in a comprehensive table. Please emphasize the significance of this study compared to theirs.
3. The title using test-retest terms seems confusing instead of attractive to readers. The reviewer suggests revising it. Moreover, it seems inspired by the title in Ref. 17. A strong, simple, and unique title is recommended.
4. The abstract style, seems more like a clinical study journal. Please revise it to the narrative or descriptive paragraph, which is more common in engineering journals such as Sensors.
Overall, the study is interesting, just the presentation style should be improved to fit the general audience of Sensors. The reviewer recommends revision to the manuscript.
Author Response
Point 1: The study presented in the submitted manuscript is interesting, the plantar pressure insole measurement in children with CP. Even though the N is relatively small.
Response 1: Thank you for your comment. We also acknowledge that the number of participants is low, but we opted to process the individuals’ lower limbs separately, raising the total sample to sixteen. Also post-hoc power analysis revealed power above 0.90 on most variables. Still our total number of participants does not significantly differ from that of other similar published studies ( Chang et al, 2015; Riad et al. 2007; Ricardo et al., 2021; Stebbins et al, 2005 , all cited in the main article).
Point 2: To fit the scope of the Journal, the reviewer suggested the manuscript should be revised because the manuscript style is more like a pre-clinical study (with a small population of the sample).
Response 2: Thank you for your suggestion. We have reviewed the overall format to better fit the style of Sensors.
Point 3: Please add a photo or schematic diagram of the measurement process, even though a commercial device may be involved in the measurement. This context will be an exciting part for most readers of Sensors. Please note that the presented photo should not disclose the patient's identity.
Response 3: Thank you for your suggestion. We have added pictures and schematic drawings of the experimental set up and pictures to illustrate the 3D plantar pressure results and measurement process. Please refer to page 4 line 126 and page 5 line 146 of the revised manuscript.
Point 4: For the comparative study mentioned in plantar measurements literature in the discussion, please summarize the results in a comprehensive table. Please emphasize the significance of this study compared to theirs.
Response 4: Thank you for your comment. We are not sure if we understood correctly your question, in particular to which comparative study you refer. Nevertheless, we have reviewed the Discussion and Conclusion sections to clarify the main results of our study. Please refer to page 11 lines 216, 228 and 239 and page 12 line 287 of the revised manuscript.
Point 5: The title using test-retest terms seems confusing instead of attractive to readers. The reviewer suggests revising it. Moreover, it seems inspired by the title in Ref. 17. A strong, simple, and unique title is recommended.
Response 5: Thank you for your suggestion. We modified the title to “Gait Analysis in children with Cerebral Palsy: Are plantar pressure insoles a reliable tool?” We believe it is more attractive now, with the focus on the plantar pressure sensors. Please refer to page 1 line 3 of the revised manuscript.
Point 6: The abstract style, seems more like a clinical study journal. Please revise it to the narrative or descriptive paragraph, which is more common in engineering journals such as Sensors.
Response 6: Thank you for your suggestion. We agree and have reviewed the abstract to better fit the style of Sensors. Please refer to page 1 line 12 of the revised manuscript.
Point 7: Overall, the study is interesting, just the presentation style should be improved to fit the general audience of Sensors. The reviewer recommends revision to the manuscript.
Response 7: We thank you for all your comments. We sincerely hope that the modifications made to our original manuscript meet the concerns you raised.

Round 2
Reviewer 2 Report
Dear Authors
The revised version is improved and give more clear view for the studies conducted. The nearly all revised parts of the manuscript are acceptable, except the important formula in line 159 in the page 5. It should defines what values are calculated i.e. ICC, therefore, the formula should start with "ICC =„ and so on. Additionally, following this formula, it is necessary to explain the variables (factors) included in the formula. For that reason it is advised to complete the text with an explanation:
,,where MSR is mean square for rows, MSE is mean square for error, MSC is mean square for columns, n is number of subjects.”
Here the questions arise:
In case of this study, what is in rows and columns and what the ,,n” is. By the way in page 6 starts from line 187, authors gives partial answer:
,,Data from each limb was processed separately (N = 16), due to the heterogeneous physical presentation of unilateral CP that composed most of the selected sample. On average we have assessed 75.8±27.9 steps on each trial.”
It is advised to relate above statement to ICC calculation according the formula presented in line 159 in the page 5.
Still the rest of formulas are not properly editing. It is advised to use a horizontal line (-) to represent the division and a period (.) to represent the multiplication. The square root of two to re-edit. The sign (x) used by the authors actually denotes other mathematical operations.
Sincerely Yours
Author Response
Response: Thank you for your comments. We have completed and edited the statistical methods section according to your suggestions. Please refer to page 5 lines 154, 157, 173 and 178. We have also reviewed the formulas in the discussion section of the article. Please refer to page 11 line 262 and page 12 line 264 of the revised manuscript. We hope that the modifications meat your concerns.
